# Structure Prediction, Evaluation, and Validation of GPR18 Lipid Receptor Using Free Programs

**DOI:** 10.3390/ijms23147917

**Published:** 2022-07-18

**Authors:** Ilona Michalik, Kamil J. Kuder, Katarzyna Kieć-Kononowicz, Jadwiga Handzlik

**Affiliations:** Department of Technology and Biotechnology of Drugs, Faculty of Pharmacy, Jagiellonian University Medical College, Medyczna 9, 30-688 Cracow, Poland; ilona.michalik@vp.pl (I.M.); mfkonono@cyf-kr.edu.pl (K.K.-K.); j.handzlik@uj.edu.pl (J.H.)

**Keywords:** GPR18, orphan GPCRs, structure prediction, homology modeling, docking studies, MD simulation, free programs

## Abstract

The GPR18 receptor, often referred to as the *N*-arachidonylglycine receptor, although assigned (along with GPR55 and GPR119) to the new class A GPCR subfamily-lipid receptors, officially still has the status of a class A GPCR orphan. While its signaling pathways and biological significance have not yet been fully elucidated, increasing evidence points to the therapeutic potential of GPR18 in relation to immune, neurodegenerative, and cancer processes to name a few. Therefore, it is necessary to understand the interactions of potential ligands with the receptor and the influence of particular structural elements on their activity. Thus, given the lack of an experimentally solved structure, the goal of the present study was to obtain a homology model of the GPR18 receptor in the inactive state, meeting all requirements in terms of protein structure quality and recognition of active ligands. To increase the reliability and precision of the predictions, different contemporary protein structure prediction methods and software were used and compared herein. To test the usability of the resulting models, we optimized and compared the selected structures followed by the assessment of the ability to recognize known, active ligands. The stability of the predicted poses was then evaluated by means of molecular dynamics simulations. On the other hand, most of the best-ranking contemporary CADD software/platforms for its full usability require rather expensive licenses. To overcome this down-to-earth obstacle, the overarching goal of these studies was to test whether it is possible to perform the thorough CADD experiments with high scientific confidence while using only license-free/academic software and online platforms. The obtained results indicate that a wide range of freely available software and/or academic licenses allow us to carry out meaningful molecular modelling/docking studies.

## 1. Introduction

G-protein-coupled receptors (GPCRs) represent the largest and most diverse group of membrane receptors. They mediate the response to extracellular stimuli by transmitting information to the cell interior. Signal transduction by GPCRs usually begins with the formation of a protein–ligand complex, which simultaneously changes the receptor conformation and activates the G protein, which triggers a further reaction cascade, a key process in human physiology. Among other things, it affects basic functions of the nervous, endocrine, and cardiovascular systems, and this in turn determines the enormous therapeutic potential of ligands for GPCRs [1]. Regardless of the adopted identification system for GPCR, receptors are subdivided into smaller subgroups within classes according to their natural ligands [2]; however, developments in molecular biology have allowed the trend to identify receptors based on the endogenous ligands to be reversed. Currently, the primary structures of GPCRs are determined by DNA sequencing techniques. This, in turn, has contributed to an imbalance between the number of receptors identified and their natural ligands. Despite significant technological advances and all the efforts of researchers, it has not yet been possible to assign the correct molecules for all GPCRs [3], namely to deorphanize them. In class A alone, the number of oGPCRs (orphan GPCRs) circles around 80.

One of these, whose gene was first located in 1997 and whose mRNA transcripts were shown to be abundantly expressed in testes and spleen, and to a lesser extent in several other tissues associated with endocrine and immune functions, is the GPR18 receptor [4]. Despite the potential natural ligands proposed—an anandamide metabolite *N*-arachidonylglycine (NAGly) [5,6,7] and the polyunsaturated fatty acid metabolite Resolvin D2 (RvD2) [8,9] (Figure 1)—GPR officially still has orphan receptor status [10] due to contradictory findings that have emerged in recent years, while lacking definitive in vivo evidence [11,12,13]. Moreover, it has been found to express constitutional activity [14,15]. However, GPR18 has been linked to the endocannabinoid system (ECS), as a number of endogenous, plant, and synthetic cannabinoids have been shown to affect its activity (eg. THC, Abn-CBD) [16,17]—though GPR18 has a low structural similarity to CB1 and CB2 cannabinoid receptors (approximately 13% and 8%, respectively). Since analogous observations have been made for two other orphan receptors—GPR55 and GPR119—it was decided to group them together within the class A lipid receptors of GPCRs [10].

The possible spectrum of therapeutic potential for GPR18 includes the treatment of immune, pain, neurodegenerative, and cancer processes, metabolism, and reduction in intraocular pressure [18,19,20].

Yet, knowing this much of a receptor, only few potent synthetic ligands have been published to date, with our group having derived the most affine first selective antagonist CB5—an imidazothiazinone derivative—followed by the slightly less potent CB27 and one order of magnitude higher affine CB148 (Figure 1), among others. This might be because, on the one hand, GPR18 is still the subject of pharmacological controversies, and on the other hand, we lack an experimentally solved structure of GPR18. Although several homology models [21,22,23,24,25] have been described in recent years (including one from our group), since then, many more accurate methods have been developed to predict and validate protein structures. This includes the thriving of methods based on artificial intelligence/machine learning (AI/ML), which were extensively used in the recent edition of the CASP and CAMEO experiments [26,27,28,29,30].

Thus, the overarching goal of the present study was to obtain a homology model of the GPR18 receptor in the inactive state, meeting all requirements in terms of protein structure quality and recognition of active ligands in virtual screening assays. To increase the reliability and precision of the predictions, different contemporary protein structure prediction methods (template-based and template-free modeling) and software were used and compared herein. In addition, we included the models generated using the competition-leaving AlphaFold2 algorithm [26] to see whether the generated models—after all the geometric and energy optimization processes—can at least marginally compare with them, both in numerical and functional estimates. To test the usability of the resulting protein models, we optimized selected structures using up-to-date protein optimization algorithms and compared them in detail on the basis of the results of the qualitative parameters of the protein structure evaluation of the ability to recognize active ligands, and calculated protein–ligand interactions.

Such selected protein models (one from each method) were subjected to docking studies with the library of known affinity antagonists. The stability of the predicted poses and intramolecular interactions of the protein itself were then evaluated by means of molecular dynamics simulations. However, these served as a basis for the overarching goal of our studies. Since most of the best-ranking contemporary CADD software/platforms require rather expensive licenses for their full usability—in addition to high CPU/GPU performance—we wanted to test whether it is possible to perform the thorough CADD experiments with high scientific confidence, using only freeware or nonpaid, academia-licensed software and online platforms, mostly with default settings.

## 2. Results

### 2.1. Selection of Generated Models

Paying attention to all numerical values, a visual assessment and an assessment of the possible functionality of individual structures—determined on the basis of the enrichment tests performed in the first step—15 homologous models of the GPR18 receptor were selected for further optimization: 3 structures from the “classical method” that exhibited the most favorable protein–ligand interactions (PY_9, PY3_10, PY3_30); 4 out of 15 models generated using the threading method (CIT_1 and IT_1–IT_3) selected on the basis of numerical scores (C-score, Ramachandran); 7 out of 10 models generated using Rosetta ab initio scripts (5 RoseTTAFold + 2 trRosetta models). Additionally, an AlphaFold2 model (DeepMind, denoted as AF-DM) GPR18 [26] was used as a reference model. However, when performing the planned calculations, a recent study by Heo and Feig was delivered [27], which suggested that AlphaFold2 might only predict one state biased towards the inactive conformation. Therefore, we decided to incorporate in this study a multistate prediction protocol that extends AlphaFold2 to predict active or inactive states with very high confidence, thus allowing the usage of two additional inactive state models: a Collab-script-generated model (denoted as CF-DM) and a model downloaded from a preprepared repository (denoted as PF-DM).

Taking into account the numerical scores, the models generated by the classical method received the weakest scores in most cases despite being subjected to the most thorough optimization process; however, this might be due to the suboptimal selection of templates, or the alignments used. On the contrary, much to our surprise, the AlphaFold2-derived/-modified models obtained generally lower protein structure qualitative estimations when compared with the remaining models as presented in Table 1, yet performed far better in the following studies. On the other hand, it often appears that experimentally solved structures also do not obtain ideal values from each of the functions used. However, given the number of variables and models generated in this study, we decided to stick to the imposed, relatively strict criteria (based on the results of estimations itself and their statistical performance) set to obtain the reliable and functional homology model of the GPR18 receptor in the inactive state.

Furthermore, only the 15 selected models that formally meet the basic qualitative and functional criteria were included for further studies, from which 5 models were finally chosen (according to the proposed “Stars” evaluation system, Table 2): PY3_10, IT_1, CIT_1, TR_1, and TTA_5. For ease of description, the model designations in the following part of the manuscript have been simplified to PY3, IT, CIT, TR, and TTA models, respectively. Adding 3 of the AlphaFold2-algorithm(s)-generated models—AF-DM, CF-CM, and PF-DM—a total of 8 different GPR18 models were used for further studies.

### 2.2. Structural Motifs Comparison

In a recent comprehensive study on molecular understanding of GPR18 [24], a group of Reggio pointed out three main structural characteristics responsible for its biological activity: the ionic lock between R119^3.50^ and S230^6.33^, the toggle switch formed by F245^6.48^, M275^7.42^, H249^6.52^, and Y104^3.35^, and the sodium-binding pocket formed by N40^1.50^, D68^2.50^, and D282^7.49^. Of all the tested models, the first condition was not met in the case of the TR and TTA models; however, these showed O-N atomic distances close to those of the H-bond (3.56 Å and 3.33 Å, respectively, and surprisingly by the AF-DM model (O-N distance of 5.11 Å). The observed ionic lock broke in all cases during the 10 ns dynamics simulation of all *apo* form protein models, which might be due to the relatively weak hydrogen bond between a charged and an uncharged partner, and is in agreement with the aforementioned studies. The changes in distances between R119^3.50^ and S230^6.33^ during the dynamic simulations for the unbound protein models described herein are presented in Figure 2.

We also checked the proposed intra-helical salt bridge interaction for the inactive states of class A GPCRs, between R3.50 and D3.49. Figure 3 illustrates plots of the heteroatom-to-heteroatom distance between R119^3.50^ and D118^3.49^ for all of the models in the *apo* forms and complexed with high-affinity CB148 ligand. This plot shows that, out of all the models, the possible salt bridge was retained only for the AF-DM, PF-DM, CF-DM, and IT models for most of the time when simulating the *apo* form. Similar behavior was observed for complexes with CB148, but with short breaks in the case of the IT model, and in the case of the PF-DM model, the interaction formed around the fourth nanosecond.

The toggle switch, formed by the residue 6.48 of the highly conserved CWXP motif (which in case of GPR18 is replaced by CFXP motif) seems difficult to control, due to the lack of a bulky residue directly across the intra-domain space, such as, e.g., F3.36 found in CB1 and CB2 [28]. In fact, when aligned to the available structures of class A GPCR lipid receptors that were bound to antagonists or inverse agonists respectively, with phenylalanine in position 6.48: CysLT1 (6RZ4, 6RZ5), CysLT2 (6RZ6-6RZ9), and PAF(5ZKQ), the conformation of F6.48 overlapped for most of the models, and likewise occurred without additional stabilization/interaction with the bulky aromatic side chain. A similar observation was found when aligned with W6.48 of the CB1 cannabinoid receptor 5UO9 PDB entry (Figure 4). However, unlike those described in [24], we did not observe the π stacking interactions between F245^6.48^ and Y104^3.35^. In most cases, F245^6.48^ was stabilized through cation–π interactions with either S278^7.45^ (IT, PY3, CF-DM), L75^2.57^ (CIT), V241^6.44^ (TR, TTA), and/or Q240^6.43^ (PF-DM). Furthermore, the position of H249^6.52^ was caged by π–π interactions with F248^6.51^ and F196^5.47^ (all 3 of the DM models) analogously as in the case of, e.g., the PAF receptor. On the other hand, the aforementioned Y104^3.35^ forms H-bonds/π interactions with D68^2.50^ and/or S278^7.45^, which are stable through the dynamic simulations. This does not apply to TR, TTA, and CF-DM, whereas Y104^3.35^ faced outward of the protein.

The choice of the template for modeling plays an important role in the further functionality of the model, as well as might bias the generated state of the model. This generally applies to the classical and threading methods also, where an inactive target is usually generated for an antagonist-bound template structure. However, this simple analogy might not be straightforward in protein folding methods. As it has been shown by recent studies of Heo [27], folding methods, with a strong indication to AlphaFold protocols, tend to show bias in the models they generate towards inactive and intermediate states vs. active states. Therefore, we incorporated the average distance measurement between Cα atoms of the last five cytoplasmic residues in TM3 (from Y120^3.51^ to V124^3.55^) and TM6 (from K227^6.30^ to I231^6.34^) as an additional activity state measure [29,30]. The values obtained from the generated models were then compared with those of active-state models either generated by CollabFold script or obtained from the repository described in [27], with average values of 15.28 Å and 15.97 Å, respectively. Figure 5 shows the changes of the average distances during the simulations for unbound and CB148-complexed protein models. In the first case, assuming the correctness of the generated active states, the TR and TTA models expressed more of an active than a nonactive state of activity (~14–16 Å) compared with the rest of the GPR18 models. They expressed a presumably inactive state with average TM3–TM6 distances in the range of approx. 10–12 Å. Similar average values were maintained for the duration of the MD simulations for most of the models with an additional increase in the average distance for an IT model. As shown in Section 2.5, the protein–ligand complexes obtained with the IT model were unstable during the simulations, which might be the basis of such a change.

### 2.3. Binding Site/Mode Comparison

Several papers have been published to date that describe the possible binding mode of GPR18 antagonists/inverse agonists. While earlier studies suggested the involvement of two arginines—R78^2.60^ and R191^5.42^ [23,31]—recent developments from Neumann [25] suggested a slightly different binding mode, involving residues of the upper parts of TM2 (R78^2.60^, Y81^2.63^, and Y82^2.64^) and TM3 (L97^3.28^) along with EL2 residues from one side and an aromatic cage formed by TM6 and TM7 amino acid side chains. Such a binding mode is common for several GPCRs [32,33], including those of the lipid ligands subfamily, for example in the platelet-activating factor (PAF) receptor structure 5ZKQ [34] (36% similarity to and 20% identity of GPR18, calculated with [2]). Nonetheless, when comparing with the same-level similarity, cysteinyl leukotriene receptor 1 structure (CysLT_1_, PDB ID: 6RZ4 [35], with bound antagonist), composed of ^2.60^RADYY^2.64^ instead of ^2.60^RMFYY^2.64^ and found in GPR18, it appears that analogical R78^2.60^ also participates in the binding of the antagonist. In fact, the involvement of these two structural areas (TM2/TM3 and TM6/TM7) might by influenced by the presence of π–bulge and π–helix, as such a correlation between protein-active sites and π–helices or π–bulges has been shown previously [36,37]. In GPR18 structure, a π–bulge of F33^1.43^ and an 8-amino-acid-long π–helix, ^5.42^RLTFFFLI^5.49^, can be found (as predicted with Quick2D [38,39]). This in turn results in (a) a bending of TM2 that determines the greater possibility of interaction of R78^2.60^, Y81^2.63^, and Y82^2.64^ with the ligand; (b) a rotation/bending of TM6 with R191^5.42^, directed inward toward the receptor. In fact, our findings were confirmed by predictions from PrankWeb [40] for all of the models. Regardless of the chosen model, a constant pattern of amino acids can be observed, which constitutes the (presumably) GPR18-binding pocket (Figure 6) with the most frequent prediction of R78^2.60^, Y81^2.63^, Y82^2.64^, T101^3.32^, C172^45.50^, L173^45.51^, K174^45.52^, R191^5.42^, F248^6.51^, F252^6.55^, L255^6.58^, N265^7.32^, G268^7.35^, A269^7.36^, and T272^7.39^.

The sizes and shapes of the predicted binding sites vary from model to model, with the largest predicted for the TTA model (130 surface atoms). The main factors responsible for this condition were the location, shape, and conformation of the EL2 amino acid side chains. In the case of IT model, EL2 formed a β-sheet hairpin (as seen, e.g., in the structure of 5ZKQ) located closer to TM3/TM4, that resulted in a wider pocket than that of the CIT model, where the loop was positioned towards TM6, therefore leaving less space for ligands to bind, as well as forcing the configuration change of R78^2.60^ (90° bend at C_δ_ downwards) and R191^5.42^ stacking parallel along TM5. EL2 of PY3 and TR models behaved similarly to that of IT, with slightly different conformations of, e.g., K174^45.52^ (in TR model facing the TM3, in PY3 upwards). Moreover, in the case of the TR model, the configuration of both arginine residues made it difficult for the ligands to interact. However, the conformation of interacting residues Y81^2.63^ and Y82^2.64^, a conformation we believe to be key, did not vary significantly for most of the models. The loop moved toward the center in the TTA model, with K174^45.52^ facing upward, and thus constituted the largest binding pocket among all models, unfortunately with no translation into the ligand-binding mode. A similar situation was observed for the AF-DM model; however, EL2 dug deepest into the cytosolic part of the receptor in this case, thus resulting in a slight bend of TM2 and a change in (presumable) binding pocket amino acids conformations, which is even more important as it hence limited the access to Y81^2.63^ and Y82^2.64^ for the ligands to bind. This resulted in a significant reduction in binding site, which was confirmed using the PrankWeb webservice—it did not detect a typical orthosteric binding pocket, but only 2–3 smaller allosteric ones. Finally, EL2 of the CF-DM model cut into the receptor with a tongue-like S167-C172^45.50^ fragment towards TM2 that caused Y81^2.63^ to bend upward, leaving very little space in the binding site.

In fact, in the case of the RoseTTAFold and DeepMind models, EL2 not only interacted with the other extracellular loops but also interacted with the side chains of TM2 amino acids indicated as crucial for the binding of selective antagonists to the GPR18 receptor, e.g.: through observed H-bonds between A170 and Y81^2.63^ and either E86^23.49^ or W87^23.50^ interactions in the case of the TTA model; through S176 and D177, forming an H-bond with R78^2.60^; and through A170-W87^23.50^, forming an H-bond for the AF-DM model. Such EL2 conformations significantly limited the ability to form favorable protein–ligand interactions during molecular docking, yet the above models scored highly in structural evaluations and exhibited features which were indicative of an inactive state.

### 2.4. Docking Comparison

For docking purposes, six structurally similar inhouse ligands with a GPR18 affinity range of 93–650 nM and inhibitory activity were used. To avoid a possible bias in ligand poses, no constraints whatsoever were used, and the receptor grids were determined solely on the basis of the predicted coordinates. As expected, shape and size differences of the binding pockets for the particular models heavily influenced both the putative binding poses and the docking scores, with the lowest calculated for IT and PY3, and the highest calculated for the DM models.

In the case of the CIT model, all ligands were packed between TM2, TM1, and TM7 in U-shaped conformations, stabilized mainly by H-bonds between imidazothiazinone carbonyl oxygen and Y81^2.63^, as well as cation interactions between K174^45.52^ and the proximal benzene ring (connected directly to thiazinone moiety, contrary to the distal terminal one(s) of the structures described herein) independently of the length of the alkyl spacer (Figure 7). Similar U-shaped ligand folding was observed for the ligands docked to the IT model; however, with the ligand core stabilized parallel to the membrane plane through the (aromatic) H-bond—formed by R191^5.42^ and F252^6.55^ with carbonyl group oxygen—proximal benzene rings placed between TM6 and TM7 and distal benzene rings (of, e.g.,: CB148) were stabilized through the cation–π interactions with K174^45.52^. In turn, despite wider binding pockets than the IT and CIT models, it was hard to find a similar binding pattern for the structurally related ligands when docked to the PY3 and TR models.

In most cases, the interactions included mostly H-bonds with both arginines but with a different structural orientation, although occupying a similar receptor area, between TM3, TM6, and TM7. However, when docking to “folded” models, the calculated binding modes started to appear more structured and meaningful, accompanied by higher and better-differentiated docking scores when compared with the models described previously. In the case of the TTA model, the imidazothiazinone head interacted with Y82^2.64^ through H-bonds with imidazole nitrogen, and formed salt bridges between carbonyl oxygen and I175 and S176. The proximal ether oxygen formed an H-bond with R78^2.60^, but the western (distal) part of the ligands was wrapped under the loop (Figure 8). For a relatively long CB92 structure, the distal benzene ring was caged by aromatic residues of TM6, which would be in agreement with the previously proposed ligand-binding pattern for GPR18 [25].

Due to the deep cutting of EL2 inside the receptor, the calculated poses for the AF-DM model differed slightly—the imidazothiazinone head was placed between TM4 and TM5, while the aromatic tail was mostly stabilized by π stacking interactions by TM6 aromatic cage amino acids, and through π–cation interactions with R191. Thus, the lower GPR18 activity of CB92 or CB5 vs. CB148 might be explained by fewer π–π interactions for this structure. However, the repository model (PF-DM) and script-generated model (CF-DM) revealed similar docking patterns for most of the tested ligands: the imidazothiazinone head fits between TM1 and TM2, interacting mainly with either Y81 or Y82, while the aliphatic–aromatic part slants toward TM5-TM6 with aromatic features stabilized by π–π interactions. Again, in the case of the PF-DM model, the poses of CB5 and CB92 did not differ much, with π stacking to an additional benzene ring of CB92. The calculated binding pose of the least affine CB27 overlaps with other ligands, yet the imidazothiazinone moiety is rotated 180° and loses contact with Y81. Yet, the longer ligands CB153 and CB154 were shown to bind conversely.

The most unified binding mode was observed for the CF-DM model, where the ligands displayed a binding mode similar to that of the PF-DM model. Distal aromatic parts were stabilized mainly by interactions with the aro–cage (F248^6.51^, H249^6.52^) with additional support from the R191^5.42^ cation. The CB148 head moiety faced and was located in very close proximity to Y82^2.64^ (distance of 2.52 Å). A similar pose was displayed by CB5, yet the lack of a second aromatic feature deprived the additional stabilization by H249^6.52^ of an aro–cage. In addition, this pose was similar to that of the PF-DM model, albeit with an additional Y82^2.64^ H-bond. Similarly, CB92 was also found to bind to CB148 and CB5.

In the next step, in order to ascertain the validity of the calculated binding poses described above and selected for the molecular dynamics simulation, we decided to use another freeware docking algorithm. As a result, we found a nearly perfect overlap of the putative binding poses which were calculated with GWO Vina for selected ligands docked to CIT, TR, TTA, and DM models. Despite these results, minor differences mostly concerning the head–tail ligand orientation were found as well. In other cases, however, ligands were calculated to dock into the same space, but no constant mode of poses or interactions could be found for structurally similar ligands. These observed differences may be due to the different search/score algorithm of GWO Vina, where both ligand and receptor conformational variables evolve iteratively. A comparison of docking modes for the same ligand/receptor pairs—using two different algorithms—is presented in Figure 9.

### 2.5. Molecular Dynamics Simulations

Molecular dynamics simulations were performed for all the proteins used, including their unbound forms as well as the complexes selected in Section 2.4. Since the docking pattern appeared more structured in the AF2-generated-models, the MD simulations of these models were performed for all of the six tested ligands. The stability of the unbound forms calculated by means of the RMSD changes when compared with the first frame (after equilibration), which is depicted in Figure 10. The values shown translate into the observed relative stability of the protein models during our short simulations, with the AF-DM model being the most stable (average RMSD of ~1.5 Å), and PY3 being the least stable (average RMSD of ~3 Å). The detailed interaction fingerprints per frame for each of the simulated complexes can be found in the Appendix A.

With respect to protein–ligand complexes, the differences in the calculated binding modes significantly influenced the behavior of the ligands. However, some regularities might be observed. With the example of the CB5 ligand, in the complexes obtained from the CIT, PY3, and TR models, the ligands were not stable during the simulations and appeared to move freely in the space between TM2, TM3, and TM7. Yet, for the CB5-TR complex, the ligand retained the starting interactions with Y81^2.63^ through the first 4 ns of the simulation, ending the simulation with a terminal benzene ring forcing TM6, with the support of π stacking from F252^6.55^. A similar observation was found for a CB5-IT complex, where the ligand initially made contacts with (mostly) EL3 and EL2 amino acid. With rather stable head moiety, the (aro)aliphatic tail movement in a semicircular manner was observed, so that it located between TM5 and TM6 in the 5th ns of the simulation. The ligand remained in this position for the entire remaining simulation time, stacked by aro–cage amino acid rings and/or R191^5.42^ that gave its cation for this purpose (Figure 11). A similar observation was made in the case of the TTA-CB5 complex. Starting from 2 ns, the ligand wrapped under the EL2 at first, changed the conformation, and around the 5–6th ns it remained stable, stacked with the F252^6.55^ aromatic ring; additionally, the formation of a supportive halogen bond with R191^5.42^ was observed. Similar observations for the (aro)aliphatic tail were found for the complex obtained with presumably the best-performing AlphaFold2-script-generated models. Despite the one-of-kind calculated binding mode of CB5 in the AF-DM model, the pose was stable through the simulation, with stable stacking of the terminal 4-chlorophenyl moiety by F252^6.55^. The imidazothiazinone head mostly interacted with L156^4.60^, Y160^4.64^, and K174^45.52^ through the whole simulation, while the proximal benzene mostly interacted with T101^3.32^, V102^3.33^, and I175 (EL2).

The most stable CB5 pose was observed for the CF-DM model, with a slight shift of the imidazothiazinone head toward TM7 within 1 ns of the simulation runtime. R191^5.42^ stabilized the distal benzene ring through cation–π interactions with additional support of F248^6.51^ and H249^6.52^ providing the stacking interactions through the whole simulation. The heterobicyclic head on the other hand made contacts with EL2 and EL3 amino acids as well, with the latter one closing over the ligand from 1 ns into the simulation. In the PF-DM-model-based complex, the conformation of the ligand changed twice during the simulation at the 1st and 5th nanosecond. However, the ligand occupied a similar space to that in the case of the CF-DM model; yet, due to slight protein structure differences, the ligand was stabilized by fewer interactions, mainly by F248^6.51^ stacking with the distal benzene ring of CB5 from one side, and by similar interactions with EL2 Y180. The overall observed changes in the ligand position and the relatively constant interactions (mostly) independently of the protein structure used in this simulation might indicate the importance of the aro–cage stabilizing interactions for antagonist/inverse agonist binding for the set of ligands used.

Similar observations were made when it came to CB148–ligand complexes. In the CB148-CIT complex, the ligand was relatively stable with constant yet very few interactions. It is worth noticing that through the whole simulation the interactions with K174^45.52^ were retained, regardless of the time of the simulation. Unlikely, in IT model-based complex, the ligand travelled from its original position, contacting mostly TM2 aminoac- ids, to the vicinity of TM5 and TM6, while staying in a U-shape conformation. Similar ligand poses and conformations of the PY3 model complex remained mostly unchanged during the simulation, yet within the duration of the simulation, they still approached TM6, so that it stacked the distal benzene ring with F252^6.55^. Similarly, for the CB148-TR model complex, the ligand conformation did not change much, yet the ligand changed the position in the receptor during the 2nd ns and halted in a similar position, with no important interactions whatsoever. Again, this changed with the “folded” models complexes. CB148 complexed with the TTA model was stably wrapped under EL2 for the whole dynamics simulation; however, it lost the initial H-bond with Y82^2.64^ in 1 ns, which was replaced by either an H-bond with Y264^7.31^ or a cation–π stabilization through R78^2.60^—this appeared occasionally during the simulation. However, the long aliphatic chain (mostly) made contacts with F248^6.51^ and F252^6.55^, and the proximal benzene ring made contacts with Y264^7.31^. Furthermore, the AF-DM-CB148 complex was stable as well, yet the head moiety rotated slowly in first 3 ns of the simulation so that the carbonyl group turned inwards toward the receptor; thus, it forced a small conformation change of the whole ligand, stabilized (most continuously) at the tail by F248^6.51^ and F252^6.55^. The head made mainly contacts with Y160^4.64^ the whole time, while the remaining part contacted and fitted in the EL2 indentation from K174^45.52^ to L181.

When it came to CF-DM, analogically to CB5, the pose of CB148 was stable; however, in the 6th ns of the simulation, the ligand dug deeper into the cytosolic part, so that R191^5.42^—instead of a cation–π interaction—provided the H-bond with a distal ether moiety, and both of the benzene rings were stacked by F248^6.51^ and H249^6.52^ and in the last 2ns by F245^6.48^ switch residue. The imidazothiazinone head retained constant Y21^1.31^, EL2, and EL3 contacts throughout the simulation. Last but not least, during the 1st ns of the simulation, the conformation of the ligand changed in the PF-DM-CB148 complex—the ligand dug deeper with its entire plane so that later in the simulation (starting from 5 ns) K174^45.52^ interacted with either the nitrogen of the head or stabilized the proximal benzene ring with cation–π; meanwhile, the distal aromatic rings were stacked by 6.51 and 6.48 phenylalanines. The CF pose however seemed more native-like. The alignment of the molecular dynamics frames for the selected ligands are depicted in Figure 12. The alignment of full-spectrum molecular dynamics frames of each ligand for each model can be found in the Appendix A.

Among the simulated complexes, the CF-DM-CB92 complex appeared the most stable, having retained most of its starting interactions. The ligand was stabilized by 3 H-bonds—between the carbonyl group and Y82^2.64^, between proximal ether oxygen and T171, and a distal H-bond with R181^5.46^. The latter replaced the initial cation–π interaction with the distal phenyl group due to delicate changes in both the ligand and the protein conformations in the 1st ns. The aromatic feature was then stabilized by π–π stacking with either H249^6.52^ or F248^6.51^ until the simulation ended. On the other hand, R78^2.60^ retained its intramolecular interaction with A110, thus stabilizing EL2. The PF-DM-CB92 complex appeared modestly less stable, as fewer strong ligand–receptor interactions were found in the complex. Due to a lack of additional stabilization, a delicate rotation of the ligands’ head appeared within 1 ns of starting the simulation, which meant that the long (aro)aliphatic part moved toward TMs and allowed for a π–cation interaction in the proximal benzene ring and K174^45.52^; additionally, it allowed π–π stacking of the distal benzene ring with F248^6.51^, and an aromatic H-bond between proximal ether oxygen and F252^6.55^. In the 6th ns, the head approached TM7, allowing for alternating contacts of the imidazothiazinone fragment with T272^7.39^ or K174^45.52^.

With regard to the CB153 CF-DM complex, in the 2nd ns, the head slowly twisted toward TM1, which allowed exchangeable H-bonds to form with Y29^1.39^; meanwhile, the terminal benzene was again stabilized by π–π with F248^6.51^ and H249^6.52^, supported by an additional cation–π interaction with R191^5.46^. The difference in affinity to GPR18 between CB92 and CB153 could be explained by the fact that, in the case of the former, R191^5.46^ interacted with an ether moiety through an H-bond. A similar interaction was found when this was compared to the most affine CB148. In fact, the nitryl moiety faced the receptor, allowing additional hydrophobic interactions to occur.

On the other hand, two possible binding modes to the PF-DM model for CB153 have been found: the “standard” mode and the inverted mode, where the head binds in the proximity of TM6. Although the inverted position would appear to be similarly stable, it appears that only the distal aromatic ring is stabilized by the K174^45.52^ cation–π and/or Y82^2.63^ π–π interactions; meanwhile, the head part was mainly held on the cation–π between the proximal ether group and F252^6.55^. Considering the standard pose, the bouquet of interactions was analogous to, e.g., CB148, with the distal part stabilized by π–π interactions with F252^6.55^, H249^6.52^, and—only in this case—Y180, while the head displayed mainly interactions with Y82 and Y264^7.31^. While comparing the “standard” pose with those obtained from CF-DM models, the head lacked the Y82^2.63^ H-bond interaction through most of the simulation. Similarly, the CF-DM-model-derived CB154 complex appeared much more stable than that of the PF-DM model, a finding supported by similar interactions to those of the CB148 complexes. A comparison of the ligand stability between both observed poses is depicted in Figure 13.

## 3. Discussion

Orphan receptor modelling, despite the increasing knowledge of possible ligand interactions with receptors, remains a challenge for modelers. Understanding the structure–activity relationship in combination with protein–ligand interactions can yield highly active and selective compounds for the desired target. In recent years, several publications describing homology modelling of the GPR18 receptor have been published, allowing an approximated understanding of possible interactions in possible orthosteric binding sites, as well as enabling descriptions of the molecular determinants of these constitutive activities. On the other hand, the last few years have seen the entry of machine learning algorithms into the modelling market, with particular emphasis on AlphaFold2 algorithms. These are set apart in the competition to such an extent that they were given a separate category in the recent CASP experiments evaluations. In this context, one may wonder why we bother to “manually” model a protein—the structure of which we could download from the Uniprot database and work on without major modifications. The answer lies in the recent literature reports on the possible bias of predicted AF2 structures. Additionally, the results obtained in this work seem to contradict this thesis. Therefore, in the present work, we decided to check whether it is possible to obtain models with parameters comparable among each other, using each of the known (more-or-less time-consuming) methods of protein generation. Moreover, in contrast to the widespread capitalization of science and monetization of results, we wanted to check whether it is possible to carry out scientific research using only commonly available software (either free or academically licensed). In other words, we wanted to determine whether it is necessary to use costly licenses to obtain reliable research results in modern science.

In order to avoid selection bias, each of the obtained models was first subjected to a detailed geometric, visual, and structural analysis in a systematic way; then, after qualifying the models for further stages (on the basis of statistical analysis), further validation procedures were imposed. Based on rather strict criteria, we selected eight structures, including three AF2-algorithm-based ones. The next step was to carefully compare the obtained structures and identify the binding sites. We first focused on the key structural elements conditioning the inactive state of the receptor and determined which of the obtained/generated models met the literature requirements for an inactive GPCR model. Our observations show that these conditions were met by most of the models tested, excluding the CIT and AF-DM models. We paid additional attention to the issue of the EL2 loop, which is likely involved in ligand binding, and which—due to the high flexibility of this fragment—we left as it was (no additional modeling procedures were implemented for the loop). Since, for the time being, the binding site of GPR18 is still conjectural, we tried to compile the available knowledge and outline a more coherent scheme of possible interactions in the proposed binding pocket itself. For this purpose we used one of the highest-ranked predictors of binding sites and, based on the aforementioned comparison, we searched for common structural features with receptors of the lipid family as well as searching for the positions of ligands in their binding pockets. On the basis of the obtained results and outcomes of the structural analyses, we predicted which amino acids might be involved in ligand binding. Our predictions have already partly coincided with published data in the literature.

The next step was to test the usability of the generated proteins on the one hand and the docking algorithms for our models on the other. However, from a theoretical point of view, the whole process of docking to different conformations of the same protein could be treated as a kind of manual ensemble docking. In order to avoid imposing docking constraints on the algorithms and thus avoid the influence of the researcher on the obtained results, the binding sites were determined on the basis of coordinates obtained from PrankWeb. This allowed the algorithms to be more independent in their search for receptor spaces. Unfortunately, in a few cases, the calculated ligand poses did not appear native-like—the algorithm rolled up relatively long ligand structures, perhaps looking for the most energetically favorable conformers. However, to prevent the introduction of possible researcher errors, in addition to visual assessment and scoring function values for all proteins, we first performed validation docking using another freeware docking algorithm. Partly to our surprise, we found that we were able to obtain complexes with similar docking poses for most models (with exception of the IT, PY3, and AF-DM models). This allowed us to select complexes for all selected models and subject them to short dynamic simulations to check the stability of the obtained complexes. Interestingly, in several cases, after short dynamics simulations, it turned out that the ligands head to the indicated location in the GPR18 receptor structure and stay there until the end (e.g., CB5-IT and CB5-TTA complexes), despite different initial putative docking poses. This may be a confirmation of our assumptions regarding the proposed ligand–protein interactions.

Without much surprise, it seems that the models obtained with the multistate AF2 algorithm were the best among those obtained; with less surprise, we observed a slightly lower performance of the “original” model from the DeepMind repository. A slight difference between ligand-binding modes and the interaction patterns resulted from the different position and degree of folding of EL2. This conclusion could be made on the basis of the following observations: (a) without precise determination of participating amino acids, but only of the binding site, the calculated binding poses showed interactions matching our findings on the one hand, and (partly) overlapping with those published so far on the other hand; (b) one can at least try to determine the influence of structural differences on binding patterns for various affine ligands; (c) the obtained poses were predicted using two different docking algorithms and were stable in molecular dynamics simulations.

Furthermore, models obtained using AF2-based algorithms significantly outperform even those models that have undergone detailed analysis and modelling on their own, despite having even lower values of numerical estimates. This then raises the following question: can we dispense of traditional methods and stay with AF2 alone? It seems to be so; however, appropriate modifications (such as “multi-state” algorithms) are necessary to generate models that reproduce the natural state as closely as possible. As with any method included in CADD, such confirmation will be obtained by obtaining empirical results.

In conclusion, taking into account all the results obtained, it can be stated that a wide range of freely available software and/or academic licenses allow us to carry out meaningful molecular modelling/docking studies. Nevertheless, the main limitation of this study was the computational power of the machines used. For the purpose of a reliable comparison between all the models obtained and the selected complexes, we were forced to perform short, 10 ns MD simulations. Reproducibility of the obtained results with the used licensed package, e.g., the most popular Schrodinger platform, as well as the generation of the active form of the protein, and the comparison of the docking/MD results between the two forms, have been chosen as subjects for further study. It should be remembered, however, that differences in results may arise due to the use of different force fields, the space-search algorithms, and the scoring functions themselves. Yet, again, ultimate confirmation can only be obtained with empirical data.

## 4. Materials and Methods

### 4.1. GPR18 Structure Prediction

In order to obtain homology models of the GPR18 receptor with a high degree of reliability, three protein structure prediction methods were used in this study, as shown in Figure 14. A total of 21 different templates were used to generate all the TBM models, which are presented, along with sequence alignments, in the Appendix A. Detailed information of FM methods templates can be found in the reference publications [27,41,42,43,44].

#### 4.1.1. Classical Method

Potential structure templates were identified using the BLAST tool implemented on the SWISS-MODEL website [45] and the NCBI database. The results obtained were then compared with data from the literature on previously published models, and the structure of the delta–opioid receptor δ (DOR) in complex with the antagonist–naltrindole (PDB ID: 4N6H; [46]) was selected as a template for the first model run. The selection was made on the basis of sequence identity, with the selected protein target at 23% (A chain) and a high protein resolution of 1.8 Å. The structure downloaded from the PDB database was subsequently preprepared using PyMOL software (Version 1.8.2.0, Schrödinger LLC, New York, NY, USA) [47] (N-terminal amino acid residues, sodium ions, water molecules, and other residues from the crystallization process were removed). Thus, the modified template was then aligned with the target sequence using Clustal Omega [48], and the resulting alignment was used for the generation of homology models of GPR18 using the PyMod 2.0 (Structural Bioinformatic Group at Sapienza University of Rome, Roma, Italy) [49] MODELLER 9.15 (University of California, San Francisco, CA, USA) [50] cap. As an output, 5 models with a bound ligand and 5 models without a bound ligand were generated.

For the generated models, appropriate side chain rotamers of the binding pocket amino acids were selected using the Dunbrack library implemented in UCSF Chimera 1.14 (Resource for Biocomputing, Visualization, and Informatics at the University of California, San Francisco, CA, USA) [51,52]. Special attention was paid to the side chains of R119^3.50^ and S230^6.33^—according to published data [13,21,53]—serving as an “ionic lock”; their interaction is crucial for protein function. The resulting structures were subjected to energy minimization using CHARMM v.c45b1 (Harvard University, Cambridge, MA, USA) [54,55,56], and protein topology and parameter files for each structure were prepared in the CHARMM-GUI [57,58,59] web interface. Two consecutive energy minimization methods were chosen: steepest descent (SD) (maximum number of minimization cycles—100) and Newton–Raphson (ABNR) (maximum number of minimization cycles—1000; tolerance to mean gradient during minimization cycle—0.01). The minimized models were visually evaluated and estimated in SAVES 6.0 [60].

For the second modeling run, based on alignments performed using PyMod (MUSCLE [61] Clustal W [62], Clustal Omega [48], and SALIGN [63]), 3 inactive-state opioid receptor templates were chosen: δ (4N6H), κ (4DJH [64]), and NOP (nociceptor; 5DHH [65]); all showed levels of sequence identity comparable to GPR18 (approximately 23%). Structure files were preprepared in PyMol (as previously stated) and the CB5 ligand was manually docked to the 4N6H structure in place of the co-crystallized ligand naltrindole. The conformation and position were chosen based on published data [25] and preliminary docking studies. The pose was validated (RMSD = 0.382) using automated protocol of the DockThor [66] server. The thus prepared template was subjected to an analogous energy minimization in CHARMM as described. A total of 40 new homology models (denoted as PY3) were constructed in PyMod: 20 with inhouse CB5 ligands (PY3_1–PY3_20) and 20 without ligands (PY3_21–PY3_40). All of them were subjected to the same preliminary qualitative analysis, and the selected structures were optimized analogously (rotamers and minimization).

#### 4.1.2. Threading Method

Protein structures were generated using the standalone version of I-TASSER 5.1 (Zhang Lab, University of Michigan, MI, USA) [67] and the web server releases I-TASSER and C-I-TASSER [68,69,70]. Based on the protein target sequence, alignments with the available template structures (from the PDB library) were performed using the LOMETS package [71,72]. The templates were automatically ranked according to the normalized Z-score (the measure of significance of the given alignment) and the level of identity of the target sequence with the given template. The “light” option (12 REMC simulations of up to 5 h each) was used to generate the standalone version models, while the other settings were left default. A total of 15 models were generated: IT_1–IT_5 (standalone I-TASSER v.5.1), IT_6–IT_10 (I-TASSER webserver, CIT_1–CIT_5 (C-I-TASSER webserver). Generated models were passed on for optimization without further proceedings.

#### 4.1.3. Ab Initio Method

The trRosetta [73,74] and RoseTTAFold [43,75] web servers (default options) were used to generate 5 models each for the target amino acid sequence, named, respectively: TR_1–TR_5 and TTA_1–TTA_5. A GPR18 receptor model generated by the DeepMind group (AlphaFold2 algorithm) was also included in the comparison and designated as AF-DM [44,76]. Due to the fact that AlphaFold2 only predicts one state that is biased depending on the templates used for model learning, additional multistate models of DM were either generated using protocol described in [27] (CF-DM) or downloaded from the repository described therein (PF-DM).

### 4.2. Preliminary Structure Evaluation

All generated models were subjected to preliminary qualitative analysis based on: (a) values determined by individual protein backbone modeling algorithms (*DOPE score* and *Objective Function Value* for MODELLER; *C-score + TM score* for I-TASSER; *TM score* for trRosetta; *Confidence* (corresponds to plDDT using DeepAccNet); *Angstroms error estimate* for RoseTTAFold; *plDDT* for AlphaFold2); (b) the SAVES6.0 Web tool estimations [60]; (c) identified primary binding site identified by the PrankWeb server [40]; (d) visual evaluation in UCSF Chimera [52] (distribution of sequences per individual transmembrane domains (TMs), positions of the most conserved amino acids, presence of H-bond between R119^3.50^ and Ser230^6.33^). Based on the above estimates, the best models from each method were selected.

### 4.3. Model Refinement

Selected models were subjected to local optimization using the 3Drefine webservice [77]. Of the 5 resulting structures, 1 for each model was selected on the basis of RWplus [78] and MolProbity scores [79]. Global refinement was then carried out for preselected models using the DeepRefiner service [80], again selecting the best model on the basis of the adopted ranking system. Due to the lower AUC values obtained for DM models after refinement with DeepRefiner in enrichment tests and, on the other hand, the best values of the initial estimates, we used the initial, non-refined structures for further studies.

### 4.4. Enrichment Test

The ligand library was generated using the publicly available ChEMBL database [81,82] (protein ID—CHEMBL2384898; structures of the ligands can be found in the Appendix A). Chemical compounds exhibiting antagonistic or inverse agonistic activity (with experimentally determined IC_50_ < 10 μM) were classified as active (26 molecules), while the rest were classified as inactive ligands (98 molecules). Three-dimensional structures were generated from SMILES strings using the OpenBabel 3.1.1 open-source tool [83]. Binding sites were determined according to the PrankWeb server predictions. Conformational analysis and ligand docking were performed using the DockThor server (binding site box size—15 Å; center coordinates from PrankWeb subjected to additional visual evaluation and possible minor modifications; docking option—standard). The receiver operating characteristic (ROC) graphs were plotted with the determination of the area under the curve values (AUC) using RStudio v1.3.1056 “Water Lily” (RStudio, Boston, MA, USA) [84], using the internal script. Enrichment was performed after the initial evaluation and optimization of the GPR18 receptor homology models (Section 4.2) and after the refinement processes described in Section 4.3. This was to test the influence of local/global optimization on possible ligand recognition ability for given protein structures. The docked positions were visually analyzed in Schrödinger Maestro. The initial and post-refinement ROC curves along with the calculated AUC values can be found in the Appendix A. In all cases (apart from the TR model—with a nonstatistical difference in the third decimal place), the AUC values were greater for the refined models on a >0.75 level.

### 4.5. Models’ Comparison

#### 4.5.1. Preliminary Ranking System

To objectively compare the generated models, a ranking system was prepared based on the results of 12 different functions that assess the global quality of protein structures. The results obtained for 73 structures were divided into 4 ranges of values based on quartiles from box plots prepared in Microsoft Excel [85] and each was assigned 0–3 stars (Table 3). For both percentage values of the Ramachandran plot (Core and Disall), a weight of 0.5 was assigned; however, after assigning the number of stars, the 2 values were summed, thus accepting the possibility of half values in the range of 0–3 stars. For labelled residues, on the other hand, the summed value of the Ramachandran plot and χ_1_–χ_2_ were considered.

#### 4.5.2. Secondary Ranking System

The sum of the stars obtained by each model in point 4.5.1 was then divided into 4 groups on the basis of analogous value distributions and again assigned the corresponding number of stars (denoted as V1; Table 4). Furthermore, the distance between the side chains of the ionic lock residues R119^3.50^ and S230^6.33^ was checked. On this basis, the V2 value also expressed in the number of stars was assigned to each structure. For the GPR18 receptor refined models (Section 4.3), the enrichment tests were performed again, and the resulting AUC values (denoted as V3) were similarly assigned a number of stars according to the criteria shown in Table 3. To preselect the best models, the V1–V3 values were summed and ranked in descending order. Based on the results, the best models were chosen for further studies. Detailed marks/rankings for each generated protein model can be found in the Appendix A.

### 4.6. GPCR Lipid Receptors Structure Comparison

The available structures of the GPCR class A lipid family receptors were obtained from the PDB database (full list can be found in the Appendix A), and preprocessed using the Schrödinger Maestro Protein Preparation Workflow with default parameters (2021.1 free-academic version, Schrödinger LLC, New York, NY, USA) [95]. The GPR18 models were aligned with protein structures using the protein structure alignment algorithm implemented in Maestro.

### 4.7. Actual Docking Procedure

Spatial structures of in-house test ligands—namely CB5, CB27, CB92, CB148, CB153, and CB154 (structures and pharmacological data of the latter two yet unpublished)—were generated analogically as described in Section 4.4. In order to test the usability of the receptors, two docking algorithms were used:
DockThor (DT) as described in Section 4.4—main docking procedure;GWO Vina 1.0 (Computational Biology and Bioinformatics Lab, University of Macau, China) [96], using mostly default parameters (except of exhaustiveness = 32)—additional, pose accuracy validation docking procedure.

The binding site determination was performed analogically as in Section 4.4. The resulting protein–ligand complexes were ranked according to the docking scores and visually checked in Schrödinger Maestro. The selected complexes (resulting from DT algorithm) were then subjected to molecular dynamics studies.

### 4.8. Molecular Dynamics (MD) Simulations

MD simulations of complexes selected in Section 4.7, as well as simulations of their *apo* forms, were performed using NAMD 2.14 for Linux (University of Illinois, Champaign, IL, USA) [97]. The positions of sodium ions were determined according to information published earlier [24]. The orientation of the protein in POPC was determined using PPM services (155 POPC molecules) and the system was prepared using the CHARMM-GUI interface [57] (ionic strength of 0.15 M NaCl; ion placement method—Monte Carlo). The water molecules were treated with the transferable intermolecular potential with a 3-point water model (TIP3P) [98]. Equilibration steps for all structures were divided into six steps using NAMD. For the first three steps, a runtime of 250 ps (picoseconds) in 1 fs (femtosecond) intervals was selected. For the last three steps, an equilibration runtime of 500 ps in 2 fs intervals was selected. The system was heated from 0 to 303.15 K during equilibration using the NPT ensemble. During the production stages, the system was kept at 303.15 K. The temperature was regulated using the Langevin dynamics thermostat. Production runs: 10 ns with 2 fs intervals. Frames were collected every 100 ps.

## Figures and Tables

**Figure 1 ijms-23-07917-f001:**
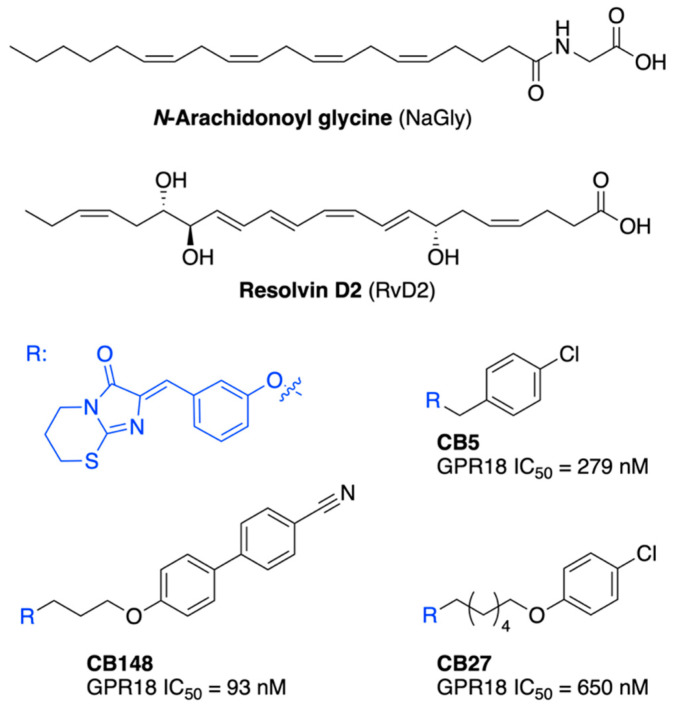
Structures of known GPR18 receptor agonists NaGly and RvD2, as well as known antagonists from our group (CB ligands).

**Figure 2 ijms-23-07917-f002:**
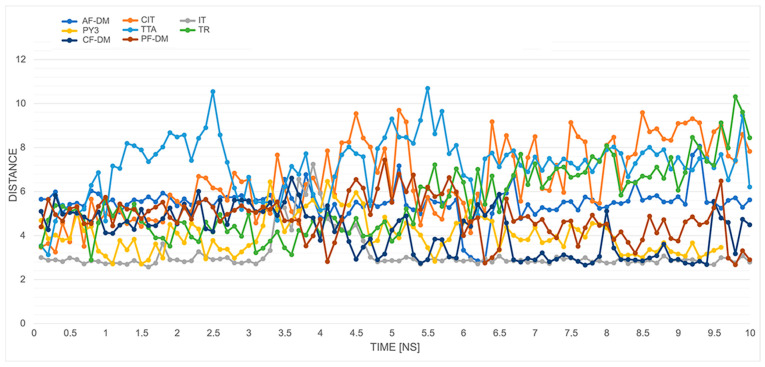
Ionic lock distances change for 8 apo forms GPR18 models during the 10 ns MD simulation. For colored graphs, please refer to the online version of this paper.

**Figure 3 ijms-23-07917-f003:**
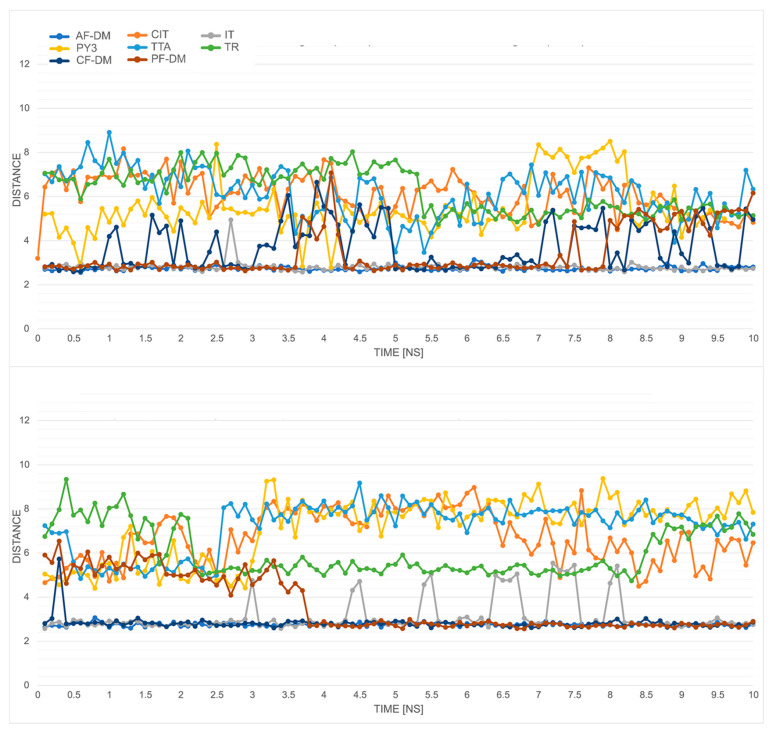
Changes in heteroatom-to-heteroatom distance between R119^3.50^ and D118^3.49^ (“aspartate cage”) for *apo* (**upper panel**) and CB148 complexes (**lower panel**) during the 10 ns MD simulations. For colored graphs, please refer to the online version of this paper.

**Figure 4 ijms-23-07917-f004:**
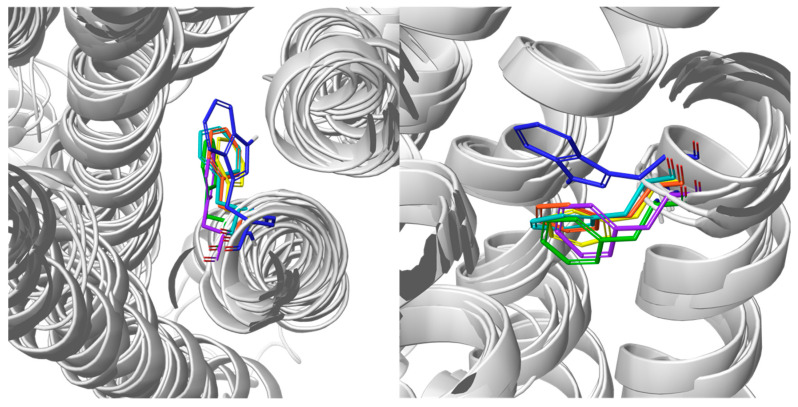
Alignment of 6.48 residues of: TTA (yellow), PF-DM (orange), and CF-DM (teal) models, and 5U09 (navy blue), 6RZ7 (violet), and 5ZKQ (green). The TM helices are represented by white ribbons for better viewing clarity. Left panel—top view; right panel—side view. For colored figures, please refer to the online version of this paper.

**Figure 5 ijms-23-07917-f005:**
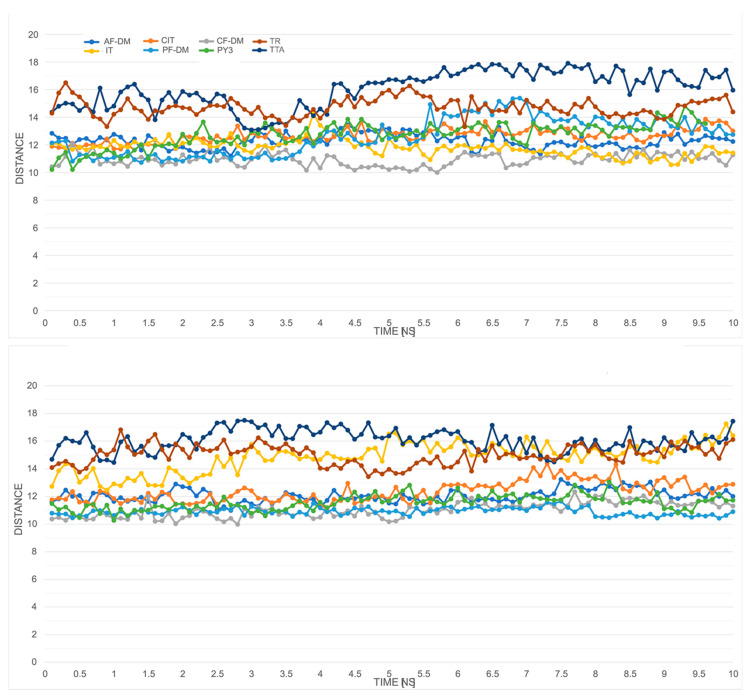
The changes in average distance measurements between Cα atoms of the last five cytoplasmic residues in TM3 (from Y3.51 to V(124)) and TM6 (from K6.30 to I6.34) for apo/unbound (**upper panel**) and CB148-complexed (**lower panel**) protein models. For colored graphs, please refer to the online version of this paper.

**Figure 6 ijms-23-07917-f006:**
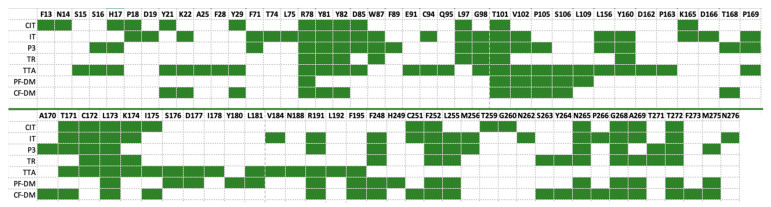
Fingerprint representation for PrankWeb binding site predictions of the models used in this study. Y axis—GPR18 models; X axis—residue with corresponding numbers. Green boxes indicate predicted residue. For a colored figure, please refer to the online version of this paper.

**Figure 7 ijms-23-07917-f007:**
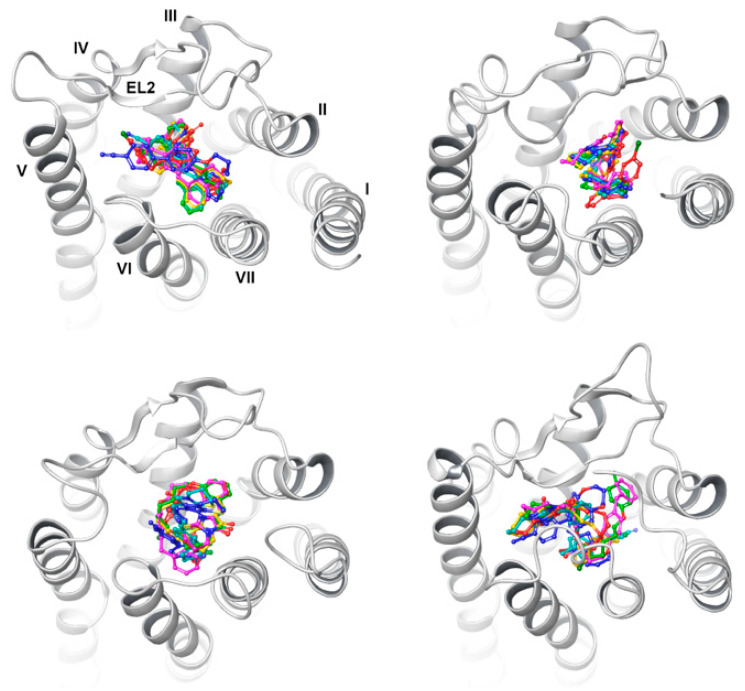
Top view of putative binding poses of the docked ligand set to IT (**top left**), CIT (**top right**), PY3 (**bottom left**), and TR (**bottom right**) models. Roman numerals indicate TMs—only marked in IT model for clarity purposes. For a colored figure, please refer to the online version of this paper.

**Figure 8 ijms-23-07917-f008:**
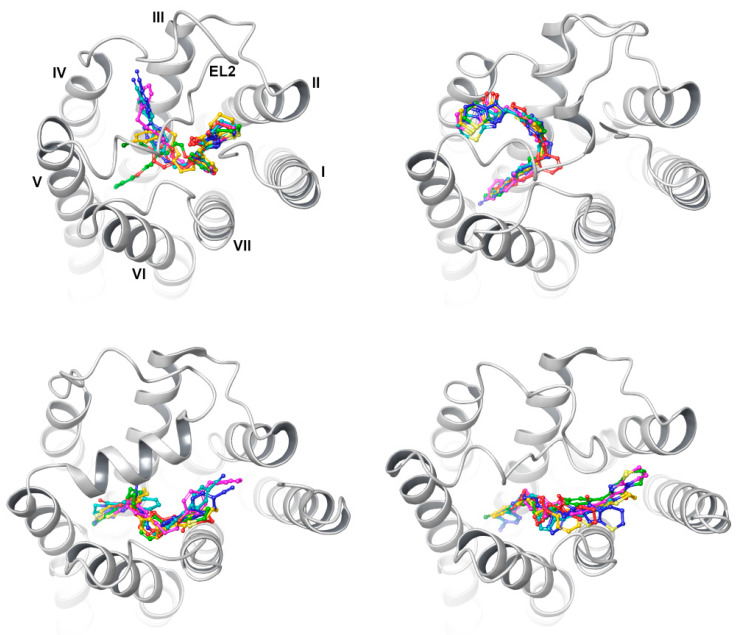
Top view of putative binding poses of the docked ligand set to TTA (**top left**), AF-DM (**top right**), PF-DM (**bottom left**), and CF-DM (**bottom right**) models. Roman numerals indicate TMs—only marked in IT model for clarity purposes. For a colored figure, please refer to the online version of this paper.

**Figure 9 ijms-23-07917-f009:**
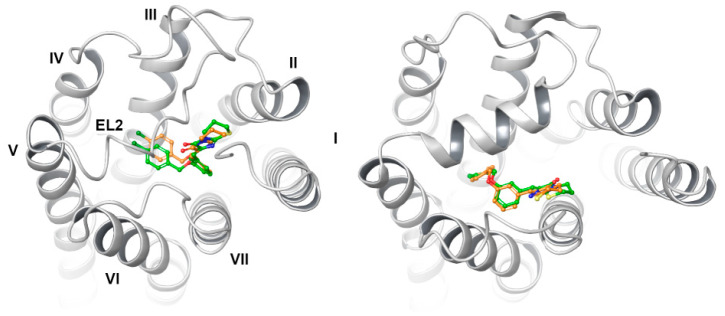
Comparison of putative binding modes of CB5 calculated with DT (green) and GWO (orange) to the TTA (**left**) and PF-DM (**right**) models. Roman numerals indicate TMs—only marked in the TTA model for clarity purposes. For a colored figure, please refer to the online version of this paper.

**Figure 10 ijms-23-07917-f010:**
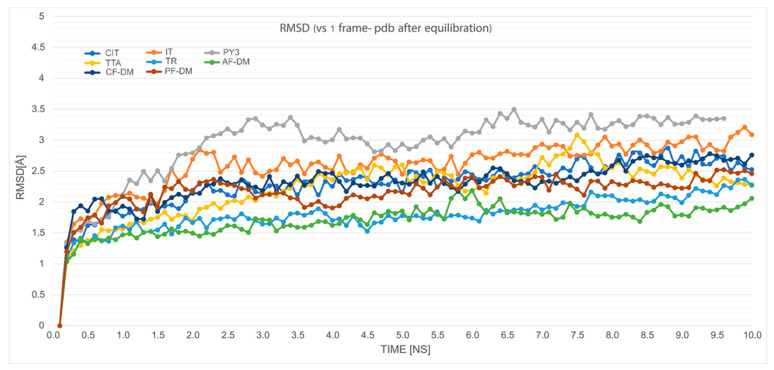
RMSD value changes measured in [Å] vs. frame 1, during 10 ns simulation of the unbound protein models described herein.

**Figure 11 ijms-23-07917-f011:**
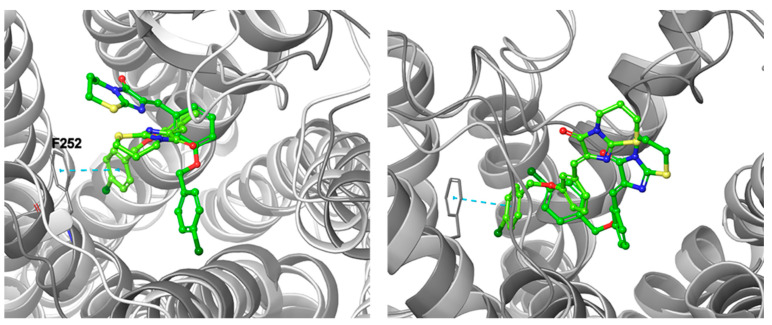
Orientations of CB5 in IT (**left panel**) and TTA (**right panel**) models at 0 ns (green) and after 10 ns of simulation (lime green). For a colored figure, please refer to the online version of this paper.

**Figure 12 ijms-23-07917-f012:**
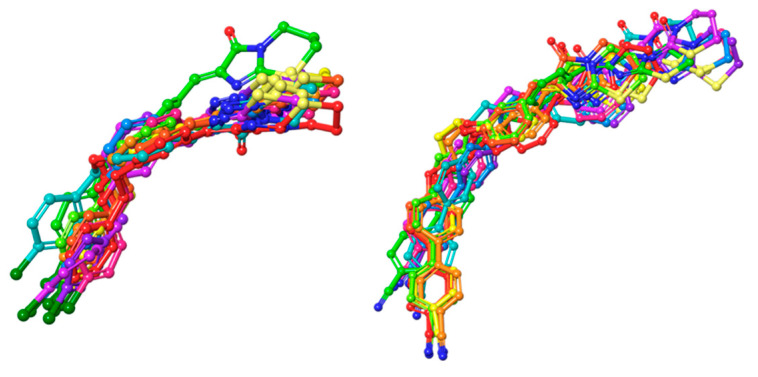
Examination of CB5 (**left**) and CB148 (**right**) compound stability in the binding pocket of the CF-DM model. Compound poses were captured at following frames: 0 ns—green; 1 ns—teal; 2 ns—blue; 3 ns—violet; 4 ns—magenta; 5 ns—pink; 6 ns—red; 7 ns—orange; 8 ns—light orange; 9 ns—yellow; 10 ns—lime. For a colored figure, please refer to the online version of this paper.

**Figure 13 ijms-23-07917-f013:**
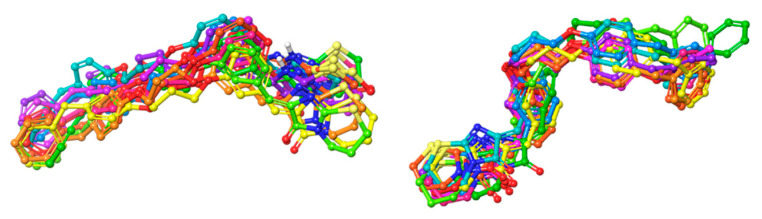
Examination of CB153 standard pose (**left**) and “inverted” (**right**) stability in the binding pocket of the PF-DM model. Compound poses were captured at following frames: 0 ns—green; 1 ns—teal; 2 ns—blue; 3 ns—violet; 4 ns—magenta; 5 ns—pink; 6 ns—red; 7 ns—orange; 8 ns—light orange; 9 ns—yellow; 10 ns—lime. For a colored figure, please refer to the online version of this paper.

**Figure 14 ijms-23-07917-f014:**
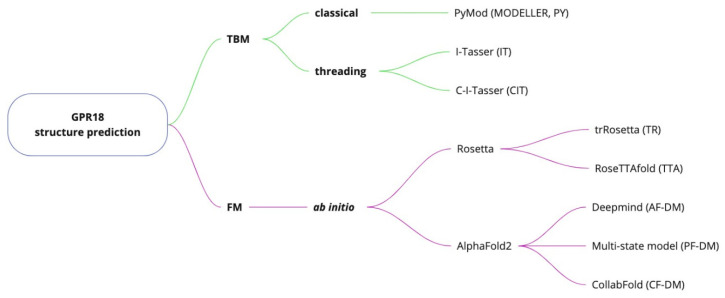
GPR18 structure prediction methods and software used in this study. Shortcuts in brackets denote particular model names prefixes.

**Table 1 ijms-23-07917-t001:** Numerical estimates of 8 the best models, out of 15 considered, including values from SAVES, 3DRefine, DeepRefiner, and AUC values from enrichment tests. Models were marked according to the software used: AF-DM—AlphaFold2; CF-DM—CollabFold script; PF-DM—multistate repository; TR—trRosetta; TTA—RoseTTAFold; CIT—C-I-Tasser; IT—I_Tasser; PY—PyMod. In each case, the N- and C-terminus were shortened (to the sequence range: 11–304) for more objective evaluation.

Model		ERRAT [%]	VERIFY 3D [%]	PROVE [%]	Ramachandran (Core, Disall [%]; Labell. Residues)			RW+ [kcal/mol]	Mol Prob	Rosetta Energy Scores	DFIRE Scores	GOAP Scores	OPUS-PSP Scores	Pred. Global Quality
AF-DM	-	99.3	53.06	2.9%	95.2	0.0	0	−7935	0.650	-	−656.4	-	-	-
CF-DM	-	88.38	52.04	4.6%	94.1	0.0	10	−7691	1.550	-	−634.9	-	-	-
PF-DM	-	95.71	44.56	5.4%	93.0	0.4	11	−7673	1.460	-	−633.5	-	-	-
TR_1	5	100.0	70.75	2.9%	95.6	0.0	1	−7987	0.880	−768.2	−654.3	−3713	−5913	0.255
TTA_5	1	98.25	60.88	2.8%	93.7	1.1	10	−7769	1.223	−829.4	−639.2	−3606	−5426	0.270
CIT_1	4	100.0	70.41	3.1%	92.6	0.4	2	−7829	1.297	−670.2	−642.0	−3487	−5876	0.237
IT_1	4	100.0	60.20	3.1%	93.3	1.1	5	−7770	1.449	−589.7	−637.5	−3434	−5678	0.214
PY3_10	1	98.60	66.33	3.6%	90.7	1.5	11	−7614	1.494	−595.3	−624.5	−3467	−5459	0.222

**Table 2 ijms-23-07917-t002:** Summary of stars ranking system of 8 models selected for further optimization: 5 models selected from preliminary ranking system (Materials and Methods, Section 4.5.1) and 3 DM models. Detailed description of V1–V3 values can be found in Materials and Methods, Section 4.5.2
*Secondary ranking system*. AUC values were obtained from the performed enrichment tests.

Model		Total Stars	V1	R119-S230 Distance [Å] ^1^	V2	V1 + V2	AUC	V3	V1 + V2 + V3
CIT_1	4	22.0	**	2.09	***	5	0.811	***	8
TTA_5	1	16.0	*	2.97	**	3	0.808	***	6
IT_1	4	11.5	-	2.41	***	3	0.797	***	6
PY3_10	1	4.0	-	1.94	***	3	0.791	***	6
TR_1	5	32.0	***	2.95	**	5	0.763	*	6
CF-DM	-	11.5	-	3.40	**	2	0.834	***	5
PF-DM	-	8.5	-	3.27	**	2	0.783	**	4
AF-DM	-	19.0	*	4.28	*	2	0.752	*	3

^1^ Distance measured between the S230 oxygen atom and the nearest R119 hydrogen atom. The distance between the S230 oxygen atom (acceptor) and the R119 nitrogen atom (donor) as well as the angle formed between the three atoms (N, H, O) were also measured, but for the purposes of the ranking system it was decided to award the corresponding number of stars based on one value only.

**Table 3 ijms-23-07917-t003:** Assigned stars value ranges of individual protein global quality functions, for the models after optimization with DeepRefiner Webserver [80].

		***	**	*	None
ERRAT [86] [%]		100	≥99	≥98	<98
VERIFY 3D [87,88] [%]		≥73	≥67	≥61	<61
PROVE [89] [%]		≤2.8	≤3.2	≤3.6	>3.6
Ramachandran plot and χ_1_–χ_2_ [90]	Core [%]; Disall [%]	≥94.4; 0.0	≥93.7; ≤0.4	≥93.0; ≤0.8	<93.0; >0.8
	Labelled residues	≤3	≤5	≤7	>7
RWplus [78]		≤−79,000	≤−78,250	≤−77,500	>−77,500
MolProbity [82]		≤1.0	≤1.2	≤1.4	>1.4
Rosetta energy scores [91]		≤−750	≤−675	≤−600	>−600
DFIRE scores [92]		≤−649	≤−641	≤−633	>−633
GOAP scores [93]		≤−37,200	≤−36,000	≤−34,800	≤−34,800
OPUS-PSP scores [94]		≤−5770	≤−5650	≤−5530	≤−5530
Predicted global quality scores [80]		≥0.26	≥0.24	≥0.22	<0.22

**Table 4 ijms-23-07917-t004:** Stars value ranges second assessment run.

	***	**	*	(None)
V1	≥27	≥20	≥13	<13
V2	≤2.5 Å	≤3.5 Å	≤4.5 Å	>4.5 Å
V3	≥0.79	≥0.77	≥0.75	<0.75

## Data Availability

Molecule structures, enrichment libraries, final protein models as well as interactive graphs of protein–ligand complexes MD simulation fingerprints, and RMSD values change are available at: https://github.com/IlonaM95/Licence-free-structure-prediction-evaluation-and-validation-on-the-example-of-GPR18-lipid-receptor (accessed 2 June 2022).

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
