# Peer review of "Structure Prediction, Evaluation, and Validation of GPR18 Lipid Receptor Using Free Programs"

_ijms, 2022, doi:10.3390/ijms23147917_

Round 1

Reviewer 1 Report

The Authors of this paper modeled the 3D structure of GPR18 receptor in their inactive state by applying different homology modelling approaches. The best models were selected according to the global quality evaluated by different on-line methods, to their capability to recognize known active compounds and to the evaluation of the complex stability by MD simulations.

In my honest opinion, it is not so important to emphasize so much the well-known role of free programs in computational researches whose quality is unquestionable and, in several cases, these programs are the state-of-art of molecular modelling. Just to cite only the most popular ones: AutoDock, Gromacs, LAMMPS, Modeller, Mopac, NAMD2, PLANTS, UCSF Chimera, VEGA, VMD, etc. I find it is exceptional nowadays that there are researches done with commercial programs only rather than free programs given the quality of the latter.

To be precise, Schrödinger Maestro, used in some part of this study (see lines 716, 749, 751, 762), cannot strictly considered a free program even if there is the “free” version only for evaluation with a very limited set of features. Why not use Chimera, VEGA, VMD, which are real free programs? In addition, Microsoft Excel is not free (line 723) as well as the operating system running Excel (Windows or MacOS).

Moreover, “Licence-free” means “without license” which is not true: freeware programs have a license, which grants you the free-of-charge use.

For the reasons exposed above, I think you should modify the title removing “Licence-free”, which is also disorienting for the reader because it is not clear what it refers to. For example: “Structure prediction, evaluation, and validation of GPR18 lipid receptor by free programs”.

I think you should indicate in the Table 1 caption the correspondence between the model labels and the used program (as shown in the main text). Moreover, you generated 15 models but in Table 1 appear only 8. You should specify also in the caption that you reported only the best model obtained by each method.

The Table 2 captions, you should report that there are 5 models selected by Stars approach and 3 modeled by AlphaFold2. Moreover, you should indicate the meaning of column labels (e.g. V1, V2, AUC) as in the Material and methods section. It is not fully clear why AlphaFold2 models are not evaluated together the others.

At line 159, it is reported Table 2 instead of Figure 2. In my opinion, you should comment the obtained results also because there are models that clearly cannot keep the ionic lock during the MD simulation.

The intra-helical salt bridge was analyzed for models with and without the CB148 ligand (as explained in the Figure 3 caption and not in the main text in which it seems that the analysis was performed only for the complexes). No comparison between models with and without CM148 is shown.

About the toggle switch: it is not clear what conformation of each model is aligned (the starting model before the MD or a MD frame). If you considered the starting model, the discussion is a little bit dangerous because the side chains are added to the backbone by the homology modelling programs by taking the conformations from a rotamer library without functional criteria, but only with energetic/statistical criteria.

The discussion about the sodium bridge between D68 and D282 as well as the distance between D282 and Q240 can be moved in the supporting information also because this paper is very long… Why did you reported the distance for only 4 models in Figure 6 instead of 8 as in Figure 3?

The movement of TM6 induces the conversion from the inactive to active state. Looking for Figure 7, it seems that TM6 (TR and TTA models) moves more when CB148, which is an agonist, is unbound. In other words, it seems that the receptor activates itself. I am expecting that the agonist can promote the activation and not the stabilization of the inactive model. What am I doing wrong?

The Figure 9 is useless, also because it is very hard to identify the EL2 domain in the models. In my opinion, the comparison of the interaction fingerprints is useless in particular if you submitted to PrankWeb the rough models before the MD. In addition, the comparison of the docking poses is not so useful. The only real information that must be extracted from the docking study is the capability to recognize between active and inactive compounds. In other words, I am speaking about ROC curve, AUC and enrichment factor, which seem calculated (as reported in tables and in the “Material and methods” section) but not discussed as expected.

Finally, you used of the full-version of Maestro to check the reproducibility of the results (lines 607-612) sounds a little bit strange, also because you used programs well known which are already fully validated.

In my opinion, this paper is too long and must be significantly shortened: 29 pages to discuss the model quality of a single protein are too much even for a very exhaustive validation. I advise you to move the less significant parts in the supporting information in order to make paper less boring and more appealing for the reader.

Author Response

The Authors of this paper modeled the 3D structure of GPR18 receptor in their inactive state by applying different homology modelling approaches. The best models were selected according to the global quality evaluated by different on-line methods, to their capability to recognize known active compounds and to the evaluation of the complex stability by MD simulations.

In my honest opinion, it is not so important to emphasize so much the well-known role of free programs in computational researches whose quality is unquestionable and, in several cases, these programs are the state-of-art of molecular modelling. Just to cite only the most popular ones: AutoDock, Gromacs, LAMMPS, Modeller, Mopac, NAMD2, PLANTS, UCSF Chimera, VEGA, VMD, etc. I find it is exceptional nowadays that there are researches done with commercial programs only rather than free programs given the quality of the latter.

To be precise, Schrödinger Maestro, used in some part of this study (see lines 716, 749, 751, 762), cannot strictly considered a free program even if there is the “free” version only for evaluation with a very limited set of features. Why not use Chimera, VEGA, VMD, which are real free programs? In addition, Microsoft Excel is not free (line 723) as well as the operating system running Excel (Windows or MacOS).

First of all, we thank very much the Reviewer 1 for such a thorough review of our manuscript and all valuable comments regarding the molecular modelling software.

 Concerning the last paragraph comments, we used Schrödinger Maestro indeed, but only the “Free Maestro” molecular visualization program (https://www.schrodinger.com/freemaestro) neither the evaluation, nor trial version. MS Word & Excel actually might not be considered as free programs, although the idea of this research was to use free CADD software, which these two do not belong to and are usually provided to the academic community for general use by the university workers/students.

Moreover, “Licence-free” means “without license” which is not true: freeware programs have a license, which grants you the free-of-charge use.

For the reasons exposed above, I think you should modify the title removing “Licence-free”, which is also disorienting for the reader because it is not clear what it refers to. For example: “Structure prediction, evaluation, and validation of GPR18 lipid receptor by free programs”.

According to the comments of Reviewer 1, the title has been updated to the proposed one

I think you should indicate in the Table 1 caption the correspondence between the model labels and the used program (as shown in the main text). Moreover, you generated 15 models but in Table 1 appear only 8. You should specify also in the caption that you reported only the best model obtained by each method.

According to the comments of Reviewer 1, the correspondence between model labels and the used program has been updated in the Table 1 caption, along with the information on the best, selected models

The Table 2 captions, you should report that there are 5 models selected by Stars approach and 3 modeled by AlphaFold2. Moreover, you should indicate the meaning of column labels (e.g. V1, V2, AUC) as in the Material and methods section. It is not fully clear why AlphaFold2 models are not evaluated together the others.

According to the comments of Reviewer 1, Table 2 caption has been updated with requested information. An explanation of the AlphaFold2 models is given in the text, in the lines 721-723.

At line 159, it is reported Table 2 instead of Figure 2. In my opinion, you should comment the obtained results also because there are models that clearly cannot keep the ionic lock during the MD simulation.

According to the comments of Reviewer 1, Figure 2 reference was placed instead of Table 2. Lines 157-157 (now 176-181) contained already the brief description of the ionic-lock breakage for the apo models. The possible reasoning for this state was updated in the text as well.

The intra-helical salt bridge was analyzed for models with and without the CB148 ligand (as explained in the Figure 3 caption and not in the main text in which it seems that the analysis was performed only for the complexes). No comparison between models with and without CM148 is shown.

This information was provided in the Manuscript, lines 164-169 (now186-190): “Figure 3 illustrates plots of the heteroatom to heteroatom distance between R1193.50 and D1183.49 for all of the models in their apo forms and complexed with high-affinity CB148 ligand. This plot shows that out of all models, the possible salt bridge was retained only for AF-DM, PF-DM, CF-DM and IT models for most of the time when simulating the apo form, and with complex with CB148 168 as well (with short breakages in case of IT model)”. The sentence has been modified for greater clarity.

About the toggle switch: it is not clear what conformation of each model is aligned (the starting model before the MD or a MD frame). If you considered the starting model, the discussion is a little bit dangerous because the side chains are added to the backbone by the homology modelling programs by taking the conformations from a rotamer library without functional criteria, but only with energetic/statistical criteria.

For the alignment purposes we used the “starting” models – after the refinement. We agree that this might be little dangerous, however all of the models were built using mostly inactive-state template structures, and the proposed rotamers might be added using energetic/statistical criteria, yet still fit to the inactive state conformations. Moreover, to our studies we added the DM models that were modeled with regards to the protein functional state with special emphasis of PF-DM and CF-DM models (Pictured on Figure 4). Therefore, we assumed that such alignment and comparison studies can be performed.

The discussion about the sodium bridge between D68 and D282 as well as the distance between D282 and Q240 can be moved in the supporting information also because this paper is very long… Why did you reported the distance for only 4 models in Figure 6 instead of 8 as in Figure 3?

According to the comments of Reviewer 1, the whole part on the sodium binding pocket has been moved to Supplementary Material. Figure 3 (now Figure 9 - Supplementary material) shows only the 4 models that maintained possible additional intra-helical H-bond between D2827.49 and Q2406.43.

The movement of TM6 induces the conversion from the inactive to active state. Looking for Figure 7, it seems that TM6 (TR and TTA models) moves more when CB148, which is an agonist, is unbound. In other words, it seems that the receptor activates itself. I am expecting that the agonist can promote the activation and not the stabilization of the inactive model. What am I doing wrong?

The CB148 ligand is an antagonist (data yet unpublished), therefore it stabilizes the inactive model.

The Figure 9 is useless, also because it is very hard to identify the EL2 domain in the models. In my opinion, the comparison of the interaction fingerprints is useless in particular if you submitted to PrankWeb the rough models before the MD.

Figure 9 has been moved to Supplementary Materials. Figure 8 does not show the interaction fingerprints – it shows the fingerprint representation instead. For each binding site residue predicted, a corresponding box is colored green. The purpose was to show the repeatability of predicted residues and possible binding residues pattern. These predictions were performed on rough models in order to determine 1) possible binding space, and it’s coordinates and 2) binding space uniformly for all of the models.

In addition, the comparison of the docking poses is not so useful. The only real information that must be extracted from the docking study is the capability to recognize between active and inactive compounds. In other words, I am speaking about ROC curve, AUC and enrichment factor, which seem calculated (as reported in tables and in the “Material and methods” section) but not discussed as expected.

According to the comments of Reviewer 1, initial and after-refinement ROC/AUC plots were placed in Supplementary Materials. Information on this was included in lines 865-869.

Finally, you used of the full-version of Maestro to check the reproducibility of the results (lines 607-612) sounds a little bit strange, also because you used programs well known which are already fully validated.

We did not use full-version of Maestro – we want to use it in the future instead, as stated in the revised manuscript lines 745-748.

In my opinion, this paper is too long and must be significantly shortened: 29 pages to discuss the model quality of a single protein are too much even for a very exhaustive validation. I advise you to move the less significant parts in the supporting information in order to make paper less boring and more appealing for the reader.

According to the comments of Reviewer 1, the manuscript has been significantly shortened. The previously indicated paragraphs / Figures of the manuscript have been moved to Supplementary Materials as requested.

Reviewer 2 Report

The manuscript “Licence-free structure prediction, evaluation, and validation on the example of GPR18 lipid receptor” is a research article that investigated the stability of the predicted poses and was then evaluated by means of molecular dynamics simulations to test the usability of the resulting models, they optimized and compared the selected structures followed by the assessment of the ability to recognize known, active ligands. And they a wide range of freely available software and/or academic licenses allows us to carry out meaningful molecular modeling/docking studies. Overall, the whole manuscript was well-organized, and the information provided in this study and the experimental methodology is interesting. Hence, I recommend its publication in IJMS after a minor revision with the following comments addressed.

minor point

  1. The conclusion looks fine, and the main limitation also should be discussed as well. Could the authors mention the limitation of the study and also what are the future plans of this work?
  2. Line 13, Don’t use “Seems” in scientific writing.
  3. There are some grammatical errors in this manuscript such as continuously forgetting to add ‘a’ or ‘the’ before a specific word which limits the clarity of the author’s writing. Check the language issues.

Author Response

First of all, we thank very much the Reveiwer 2 for review of our manuscript and all valuable comments.

  1. The conclusion looks fine, and the main limitation also should be discussed as well. Could the authors mention the limitation of the study and also what are the future plans of this work? –

The main limitations as well as future plans have been added to the Conclusion section (lines: 607-613).

  1. Line 13, Don’t use “Seems” in scientific writing.

This issue has been corrected in the reviewed version of the manuscript.

  1. There are some grammatical errors in this manuscript such as continuously forgetting to add ‘a’ or ‘the’ before a specific word which limits the clarity of the author’s writing. Check the language issues.

The language has been double checked before submitting the manuscript with DeepL translator (UK English) as well as Open Writefull app and all missing ‘as’ and ‘the’ have been added to the present version of the manuscript.

Reviewer 3 Report

The authors developped several structural models of the GPR18 receptor and investigate the binding of some ligands to the models. The work was peformed with “free” tools. In the discussion they compare and contrast the use of different “homology” modeling strategies.
The paper can be interesting for people working in the field of GPCR research.

Minor
I do not understand this:
ionic lock between R119 3.50 and S230 6.33
Ionic interactions are between charged atoms ? Serine is not charged ? maybe hydrogen bond? Please clarify

Author Response

First of all, we thank very much the Reviewer 3 for a thorough review of our manuscript and all valuable comments .

Minor
I do not understand this: ionic lock between R119 3.50 and S230 6.33 Ionic interactions are between charged atoms? Serine is not charged ? maybe hydrogen bond? Please clarify

Indeed, the “ionic lock” usually is an interaction between an arginine in TM3 (R3.50) and a negatively charged residue in TM6 (D/E6.30). However, GPR18 is a special receptor in many aspects. While GPR18 has an arginine at 3.50, it lacks a negatively charged residue at the end of TMH6. Instead, GPR18 has a hydrogen bonding residue, S6.33, at this position that interacts with R3.50 (source: Int J Mol Sci. 2019 May; 20(9): 2300, referenced in Manuscript.

Round 2

Reviewer 1 Report

The Authors modified significantly the paper and addressed all my comments even if a minor issue remains. In particular, CB148 is shown in Figure 1 whose captions is “Structures of known GPR18 receptor agonists…” while the Authors replied me that is an antagonist and in this way its behavior is well justified in Figure 7 (now 5). I think you must change this caption, also because as reported in Biomolecules 2020, 10, 686; doi:10.3390/biom10050686, CB5, CB27 and CB148 are antagonists.

Author Response

The Authors modified significantly the paper and addressed all my comments even if a minor issue remains. In particular, CB148 is shown in Figure 1 whose captions is “Structures of known GPR18 receptor agonists…” while the Authors replied me that is an antagonist and in this way its behavior is well justified in Figure 7 (now 5).

We would like to thank for the review of our revised work. Indeed, the caption of Figure 1 may have been misleading and has therefore been corrected as suggested by the reviewer.